# Isosteviol Sodium (STVNA) Reduces Pro-Inflammatory Cytokine IL-6 and GM-CSF in an In Vitro Murine Stroke Model of the Blood–Brain Barrier (BBB)

**DOI:** 10.3390/pharmaceutics14091753

**Published:** 2022-08-23

**Authors:** Moritz Reschke, Ellaine Salvador, Nicolas Schlegel, Malgorzata Burek, Srikanth Karnati, Christian Wunder, Carola Y. Förster

**Affiliations:** 1Department of Anaesthesiology, Intensive Care, Emergency and Pain Medicine, University Hospital Würzburg, 97080 Würzburg, Germany; 2Section Experimental Neurosurgery, Department of Neurosurgery, University Hospital Würzburg, 97080 Würzburg, Germany; 3Department of General, Visceral, Transplant, Vascular and Pediatric Surgery, University of Würzburg, 97080 Würzburg, Germany; 4Institute of Anatomy and Cell Biology, University of Würzburg, 97070 Würzburg, Germany; 5Department of Anesthesia and Intensive Care Medicine, Robert-Bosch Hospital, 70376 Stuttgart, Germany

**Keywords:** IL-6, ischemia, isosteviol sodium (STVNA), dexamethasone, glucocorticoid receptor, cerebEND

## Abstract

Early treatment with glucocorticoids could help reduce both cytotoxic and vasogenic edema, leading to improved clinical outcome after stroke. In our previous study, isosteviol sodium (STVNA) demonstrated neuroprotective effects in an in vitro stroke model, which utilizes oxygen-glucose deprivation (OGD). Herein, we tested the hypothesis that STVNA can activate glucocorticoid receptor (GR) transcriptional activity in brain microvascular endothelial cells (BMECs) as previously published for T cells. STVNA exhibited no effects on transcriptional activation of the glucocorticoid receptor, contrary to previous reports in Jurkat cells. However, similar to dexamethasone, STVNA inhibited inflammatory marker IL-6 as well as granulocyte-macrophage colony-stimulating factor (GM-CSF) secretion. Based on these results, STVNA proves to be beneficial as a possible prevention and treatment modality for brain ischemia-reperfusion injury-induced blood–brain barrier (BBB) dysfunction.

## 1. Introduction

Among the major causes of death and morbidity worldwide, ischemic stroke remains to be the most prevalent, growing continuously due to the aging of the population [1]. Stroke can be defined as an acute neurological dysfunction caused by the interruption to the vasculature supplying the brain [2]. Ischemia involves various neuroinflammatory cascades leading to a series of events including blood–brain barrier (BBB) breakdown [3,4]. BBB disruption due to ischemic stroke aggravates brain injury and cognitive impairment [5]. This happens when neuroinflammation accompanied by the release of cytokines, chemokines, metalloproteinase and vascular endothelial growth factor (VEGF) ensues following stroke, leading to infiltration of inflammatory cells, which compromises the BBB [6]. These sequelae of events give rise to vasogenic edema, hemorrhagic transformation and increased patient mortality [7,8]. Furthermore, BBB disruption has been shown to be a marker for stroke severity and a negative predictor for long-term outcome after stroke [9].

As such, it is important to be able to discover and develop new therapeutic modalities to target stroke-induced BBB dysfunction. One approach is through the reduction of inflammatory mediators that induce hyperpermeability. This can be achieved, for instance, through the administration of glucocorticoids (GC), which are known to tighten the BBB [10]. One such glucocorticoid is dexamethasone. The action of dexamethasone, which includes reduction of inflammation and edema after cerebral hemorrhage and ischaemic damage [11], is dependent on glucocorticoid receptor (GR) binding [12]. Although clinical studies could not demonstrate benefits of dexamethasone on patient outcome [13,14,15], it was reported that dexamethasone, in combination with the proteasome inhibitor bortezomib, reduced BBB permeability and brain edema [16]. After traumatic brain injury, glucocorticoid insensitivity occurs at the hypoxic BBB. This is due to the post-translational GR modification that takes place whereby degradation of proteosomal GR ensues in brain endothelial cells. Inhibition of proteosomal GR degradation results in decreased post-traumatic cerebral edema formation and attenuated neuronal injury [16,17].

Recently, we demonstrated the neuroprotective effects of isosteviol sodium (STVNA) to murine brain capillary cerebellar endothelial cells after hypoxia in vitro. In this study, a significant reduction in cell volume was observed with the simultaneous administration of STVNA and oxygen glucose deprivation (OGD). Moreover, expression of barrier proteins, as well as integrin-α, which are significant regulators of BBB integrity and downregulated by OGD, increased with STVNA treatment [18]. STVNA is a product of stevioside acid hydrolysis, which demonstrated anti-oxidative, anti-inflammatory and anti-apoptotic effects in several studies [19,20,21]. As such, it is a promising therapeutic modality whose outcome warrant further investigation. A study reported that stevia compounds demonstrate GR-modulatory activity in a cell-type specific manner. They evaluated the impact of steviol, steviol glycoside and a stevia extract on the GR signaling cascade in Jurkat leukemia cells, as well as peripheral blood mononuclear cells (PBMCs). The stevia compounds illustrated GR-modulatory activity in T-cells but not in PBMCs [22].

In this current study, we tested the hypothesis that STVNA can activate GR transcriptional activity in brain endothelial cells as previously published for T cells. Furthermore, we examined the effects of STVNA to inflammatory cytokine secretion in our in vitro BBB model. Since STVNA manifests potential as supplemental modality to alleviate stroke-induced BBB dysfunction leading to poor patient outcome, the results we present herein add to the presently available baseline knowledge on STVNA as a neuroprotectant.

## 2. Materials and Methods

### 2.1. Chemicals

Dexamethasone was purchased from Sigma-Aldrich. Isosteviol Sodium was synthesized as described previously [18].

### 2.2. Cell Culture

Murine microvascular brain endothelial cells (cerebEND) [23,24,25,26] were grown in Dulbecco’s Modified Eagle’s Medium (DMEM) supplied with 1% penicillin/streptomycin and 10% fetal calf serum (FCS) on gelatin-coated 6- or 96-well plates (Thermo Fisher Scientific, Waltham, MA, USA). They were cultured in a 37 °C incubator (Heracell VIOS 250i, Thermo Fisher Scientific, USA) until 95% confluent as observed under the microscope.

### 2.3. Oxygen-Glucose Deprivation (OGD) and Isosteviol Sodium (STVNA) Treatment

Prior to treatment, the cells were differentiated by growth in DMEM medium containing 1% charcoal-stripped FCS for 24 h. Pre-treatment with STVNA followed by OGD or post-OGD treatment with STVNA were selected for the experiments. Pre-treatment groups were treated with 10/20/30 mg/L STVNA and/or 100 nMol Dex for 24 h followed by 4 h-OGD and 20 h of post-OGD treatment. Post-OGD treatment with STVNA was carried out with 10/20/30 mg/L STVNA and/or 100 nMol Dex for 20 h in DMEM containing 1% FCS. OGD was performed in glucose- and glutamine-free DMEM (for 4 h with 37 °C, 5% CO_2_ and 1% O_2_ in an OGD-incubator (Heracell 150i, Thermo Fisher Scientific) as described previously [27,28,29]. The various concentrations of STVNA were obtained by weight (salt) per volume (medium) dilutions.

### 2.4. Western Blot

Western blot was performed as previously described [30,31]. Briefly, cells were washed twice with cold phosphate buffered saline (PBS) and scraped off with RIPA buffer (50 mM Tris pH 8.0, 150 mM NaCl, 0.1% SDS, 0.5% sodium deoxycholate, 1% NP40) containing protease inhibitor cocktail (Roche, Basel, Switzerland). Next, cells were sonicated (Bandelin SONOPULS, Germany), diluted for equal protein content and mixed with Laemmli buffer containing 5% β-mercaptoethanol (Sigma-Aldrich, St. Louis, MO, USA). After denaturation, samples were run through a 10% SDS- PAGE mini gel and blotted overnight using a Mini Trans-Blot Electrophoretic Transfer Cell (BioRad, Feldkirchen, Germany). Subsequently, the membrane was blocked in 5% non-fat dry milk (Carl Roth, Karlsruhe, Germany) and probed with the primary antibodies against glucocorticoid receptor (1:200, Santa Cruz, #sc-8992; 1:500, Santa Cruz, #sc-393232), lamin A (1:500, Abcam, #ab8980) and tubulin (1:10000, Abcam, #ab197740), followed by secondary antibodies anti-mouse/rabbit (1:3000, Cell Signaling Technology, #7074/#7076). Detection was carried out using an enhanced chemiluminescence (ECL) solution and viewed with Imagen Flour Chem FC2 (Cell Biosciences) and the AlphaView Software (Version 1.3.0.7, Innovatech Corporation, Troisdorf, Germany). Densitometric analyses were performed using ImageJ software (NIH and LOCI, University of Wisconsin, Madison, WI, USA).

### 2.5. Nucleus/Cytoplasm Fractionation

Nuclear and cytoplasmatic extraction was performed using the NE-PER Nuclear and Cytoplasmatic Extraction Kit (Thermo Fisher Scientific), following the manufacturer’s instructions. The cell lysates were subjected to SDS-PAGE and analyzed by Western blot as described above.

### 2.6. Immunofluorescence Staining

Cells were cultured on Collagen IV (Sigma Aldrich)-coated cover slips. After treatment, they were washed twice with PBS, fixed with −20 °C methanol for 10 min and then again washed twice with PBS. Next, they were blocked in 5% normal porcine serum for 1 h, before being stained with antibodies against glucocorticoid receptor (1:100, Santa Cruz, #sc-8992) and claudin-5 Alexa Fluor (1:2000, Thermo Fisher Scientific, #352588) at 4 °C for 3 days. Secondary anti-rabbit Alexa Fluor antibodies (1:2000, Thermo Fisher Scientific, #A21207) and DAPI (Sigma Aldrich) were added at room temperature for 1 h. The cover slips were washed with PBS three times, and mounted on microscope slides using PermaFluor mounting (Thermo Fisher Scientific). Visualization was performed using a confocal microscope with 60× magnification (Nikon Eclipse TE2000-E, Tokyo, Japan).

### 2.7. Multiplex ELISA

Analysis of cytokine and chemokine biomarker levels in the medium of treated cells was done using the Mouse High Sensitivity T Cell Magnetic Bead Panel 96-Well Plate Assay (Merck Millipore, Cat. No. MHSTCMAG-70K)) for the following analytes: GM-CSF (granulocyte-macrophage colony-stimulating factor), IFNg (interferon gamma), IL (interleukin)-1a, IL-1beta, IL-2, IL-4, IL-5, IL-6, IL-7, IL-10, IL-12, IL-13, IL-17A, KC (keratinocyte-derived chemokine), LIX (LPS-inducible chemokine), MCP (monocyte chemoattractant protein)-1, MIP (macrophage-inflammatory protein)-2 and TNFa (tumor necrotic factor a), following the manufacturer’s instructions.

### 2.8. Transfection and Luciferase Assay

Transfection and luciferase assays were performed as previously described [31,32]. First, plasmid DNA was amplified by transformation of Library Efficiency DH5α competent cells (Thermo Fisher Scientific). A pGL3 Firefly Luciferase Reporter Vector (Promega, Madison, WI, USA) containing glucocorticoid-responsive test promoter pMMTV-DNA [17] and a Renilla Luciferase Control Reporter Vector (Promega) were purified using the innuPREP Plasmid Mini Kit 2.0 (AJ Innuscreen, Berlin, Germany). The amount of extracted vector was determined using a NanoDrop 2000 instrument (Thermo Fisher Scientific). Next, cerebEND cells were co-transfected with both vectors using the Lipofectamine 3000 Reagent Kit (Thermo Fisher Scientific). MMTV transcription activity was measured using the Dual-Luciferase Reporter Assay System (Promega) by determining the firefly luciferase signal in a Lumat LB9507 (Berthold Technologies) and normalizing it to transfection efficiency by determining the Renilla luciferase activity.

### 2.9. Statistical Analysis

Each experiment was repeated two to five times. Data were analyzed using one-way ANOVA with post hoc Tukey’s multiple comparison test, and *p* < 0.05 was considered significant. Statistical analyses were conducted using the GraphPad PRISM 7 (GraphPad Software, Inc., San Diego, CA, USA) software.

## 3. Results

### 3.1. STVNA Administration Demonstrates a Trend towards Reduced OGD-Induced IL-6 Expression

Oxygen glucose deprivation (OGD) of cells in culture, a widely accepted in vitro model of ischemia, implicates an activated inflammatory cascade. Specifically, it increased IL-6 levels in our murine microvascular cerebral endothelial cells cEND [29]. Hence, in this study, we tested if isosteviol sodium (STVNA) would demonstrate an ability to affect the level of IL-6 secretion in relation to OGD. We tested cell cultures, each of which was divided into four treatment groups—pre-normoxia, post-normoxia, pre-OGD and post-OGD. Indeed, we could detect a trend towards reduced IL-6 expression in our murine microvascular cerebellar endothelial cells cerebEND upon administration of STVNA post-OGD in comparison to the control. The same effect was shown by 100 nmol dexamethasone, a known potent anti-inflammatory therapeutic agent. Likewise, the combination of 10/20/30 mg/L STVNA + 100 nmol dexamethasone exhibited a decreased IL-6 level (Figure 1).

OGD resulted in increased inflammation in microvascular endothelial cells in vitro [29]. The administration of 20 mg/L STVNA decreased IL-6 secretion, both alone and in combination with dexamethasone. Since most effects were found post-OGD, further investigations were conducted under this condition.

### 3.2. Glucocorticoid Receptor (GR) Activity Is Unaltered by STVNA

The glucocorticoid receptor (GR) arbitrates the actions of glucocorticoids in cells [33,34]. As previously published for Jurkat cells, STVNA led to transcriptional activation of GR [22]. We also considered the effects of STVNA on GR through Western blot, immunofluorescence staining and luciferase assays. In the Western blot of cytoplasmic and nuclear fractions, STVNA treatment resulted in nuclear translocation of GR only when dexamethasone was present. No effects with STVNA alone were observed. (Figure 2A).

Western blot assay of whole lysates, on the other hand, showed significant decrease in GR expression (*p* ≤ 0.0001) upon treatment with both 100 nmol dexamethasone and 20 mg/L STVNA in both normoxic and post-OGD groups, similar to 100 nmol dexamethasone alone. Treatment with 20 mg/L STVNA alone did not affect GR expression (Figure 2B). Furthermore, confocal immunofluorescence microscopy revealed a slight, albeit indefinite, nuclear translocation of GR resulted as an effect of STVNA treatment as observed in the cells treated with 20 mg/L STVNA post-OGD (Figure 2C).

The GR is a ligand-activated transcription factor that controls gene function in many physiological processes. It binds to target sites within promoter regions of genes assembled as chromatin, leading to changes in nucleosomal architecture [35]. As a well-established model to investigate GR transcriptional activation, we used the mouse mammary tumor virus (MMTV) promoter to determine whether STVNA can activate the transcriptional activity of GR. cerebEND cells were transfected with the promoter–reporter construct pMMTV-luc and assayed for luciferase activity [34,36]. Treatments with 100 nmol dexamethasone alone or in combination with 10/20/30 mg/L STVNA were significantly different to control, in both the normoxia and OGD groups. Treatment with STVNA alone had no effects on GR-transcriptional activity (Figure 3). This posits that there is no transactivation of GR-responsive element after treatment with STVNA.

### 3.3. Decreased IL-6 and GSM-CSF Expression after STVNA Treatment

Upon measuring a panel of 18 T-cell cytokines in cell culture medium, we found that it is mostly dexamethasone that led to reduced levels of pro-inflammatory cytokines. IFNg, IL-1beta, Il-7, IL-12 and TNFa were under the detection level. MIP-2 and KC significantly increased, while MCP-1 and IL-5 decreased, with dexamethasone in combination with OGD. Nonetheless, no difference in their expression among those treated with OGD alone versus OGD in combination with STVNA was observed. Meanwhile, in the case of GSM-CSF and IL-6, a strong and significant reduction was observed upon treatment with 20 mg/mL STVNA after OGD (Figure 4). This reduction was as strong as that of treatment with dexamethasone, and to the authors’ best knowledge, was demonstrated for the first time for STVNA. This further confirms, for the first time, decreased IL-6 expression being an effect of STVNA administration.

## 4. Discussion

We recently demonstrated the neuroprotective effects of STVNA after hypoxia in our in vitro stroke model [18]. In addition, it has been shown that stevia compounds impart GR-modulatory activity in Jurkat cells [22]. To further explore its effects, we investigated the GR-modulatory activity of STVNA in our model. Moreover, we also looked at its effects on inflammatory response.

Acute inflammation accompanies the early stages of stroke, leading to detrimental side effects. Therefore, any treatment modality should be able to address it. High IL-6 levels showed to be a negative prognostic marker for the 90-day mortality after stroke [37]. In our current study, STVNA administration lowered IL-6 levels after oxygen-glucose deprivation (OGD) treatment. As such, the prospect of STVNA as a treatment strategy to lower IL-6 levels in the advent of stroke is supported. The ability of STVNA to reduce IL-6 levels was already displayed in neuroblastoma cells [38]. Its capability to produce the same effects in cerebellar endothelial cells indicates the feasibility of it reducing the negative effects associated with BBB disruption during brain ischemia-reperfusion injury.

Cerebral ischemia can disrupt the balance between pro- and anti-inflammatory responses in the brain. In our BBB model, we found no dose dependent effect of STVNA on IL-6 levels after OGD in cerebEND, since it was only with 20 mg/L STVNA that a reduction in IL-6 was detectable. IL-6 is marginally expressed in normal brain tissue, but significantly elevated in response to injury and stroke [39]. During cerebral ischemia, IL-6 is produced by several sources: neurons, oligodendrocytes, astrocytes and vascular endothelial cells [40]. However, during cerebral ischemia, microglia are the key cells in the pro-inflammatory reaction in the brain [41]. After vascular occlusion, IL-1 and IL-6 expression increases, and they act on vascular endothelial cells to express ICAM-1 and selectins, causing leukocyte aggregation and adhesion, and mediating the inflammatory cascade to worsen cerebral ischemic injury [42]. It has been shown recently that pretreatment with a Jak2 inhibitor (AG490) in a rodent stroke model alleviated brain endothelial cell barrier disruption and decreased the reduction of tight junction ZO-1 protein caused by IL-6 [43]. The effect of IL-6 on brain endothelial cells in the context of stroke and ischemia-reperfusion injury remains clear; however, the question of whether brain endothelial cells are a major source of IL-6 in this context remains unclear. Therefore, our model is limited to demonstrating for the first time the effect of STVNA on IL-6 production in cerebEND, an endothelial cell line. With our findings in this paper, 20 mg/L shows to be the effective concentration to reduce IL-6 after OGD in this specific cell population.

Pre-treatment with glucocorticoids such as dexamethasone reduces brain edema during the early stages of ischemia [44]. In addition, they are also used in reducing tumor-associated edema [45]. These effects are modulated by GR-activation, as most responses to glucocorticoids at the cellular level are mediated by the GR. IL-6 production is noted to be repressed by GR activation [46]. Additionally, it has recently been shown in humans that glucocorticoid resistance is associated with poor functional outcome after stroke [47]. In our study, although we observed IL-6 downregulation after STVNA treatment, STVNA did not affect glucocorticoid receptor (GR) activity. Even though we detected a slight nuclear translocation of the GR after treatment with 20 mg/L STVNA in the immunofluorescence images, the activation of the GR was ruled out by the negative results in the luciferase assay. Hence, this points out that STVNA-induced repression of IL-6 is not due to a GR activation-dependent pathway. The stabilizing effect of glucocorticoids on the BBB is hampered after proteosomal GR degradation, which results in cerebral lesions [16]. Therefore, a substance that is not dependent on GR activity, such as STVNA, might be an advantageous therapeutic option. In addition, we showed, for the first time, that the mechanism described for Jurkat cells does not work in brain microvascular endothelial cells, again pointing out that the effects of GR are tissue-specific.

In addition to lowering IL-6 expression, STVNA treatment was also shown to lower granulocyte-macrophage colony stimulating factor (GM-CSF) post-OGD. GM-CSF is a proinflammatory cytokine with neuroprotective and angiogenic properties exhibited during in vivo models of cerebral ischemia [48]. In vitro, GM-CSF promotes macrophage, neutrophil and eosinophil survival and activation [49]. Of note, GM-CSF was shown to promote arteriogenesis in cardiovascular ischemic disease, thereby ameliorating brain damage [50]. Among stroke patients, it was observed that GM-CSF levels were higher compared to healthy controls. Nonetheless, no relation was found with a better neurological outcome [48]. On the other hand, high systemic levels of GM-CSF after stroke reduced stroke-induced immunodepression and post-stroke pneumonia [51]. As such, the role of GM-CSF needs to be further investigated. If lower GM-CSF would prove to be beneficial in leading to better outcome among stroke patients, STVNA treatment could be a targeted strategy to lower GM-CSF levels.

This study validates the tissue-specific effects of GR. By and large, STVNA could be an interesting treatment option due to the effects it is able to generate independent of GR. Notwithstanding, similar to any other treatment, STVNA demonstrates certain limitations as a potential treatment due to lack of efficacy on most of the chemokines and cytokines tested herein. It is known that the effects of dexamethasone are broad. Still, our present study suggests significant benefits of STVNA treatment, specifically in reducing IL-6 and GM-CSF expression at the cellular level, using our in vitro model of stroke, which signals the need for further investigation, by including, for instance, permeability assays.

## Figures and Tables

**Figure 1 pharmaceutics-14-01753-f001:**
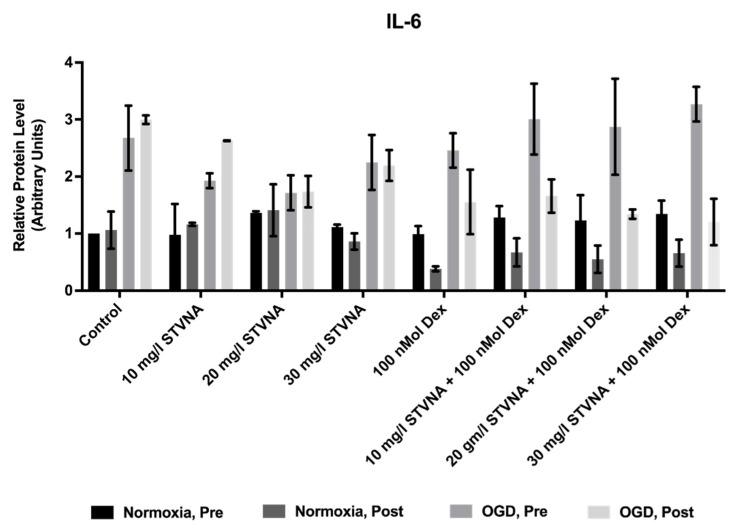
Secretion of pro-inflammatory cytokine IL-6. Multiplex ELISA reveals a trend towards reduced IL-6 expression upon administration of 20 mg/L STVNA, as well as STVNA in combination with 100 nMol dexamethasone post-OGD.

**Figure 2 pharmaceutics-14-01753-f002:**
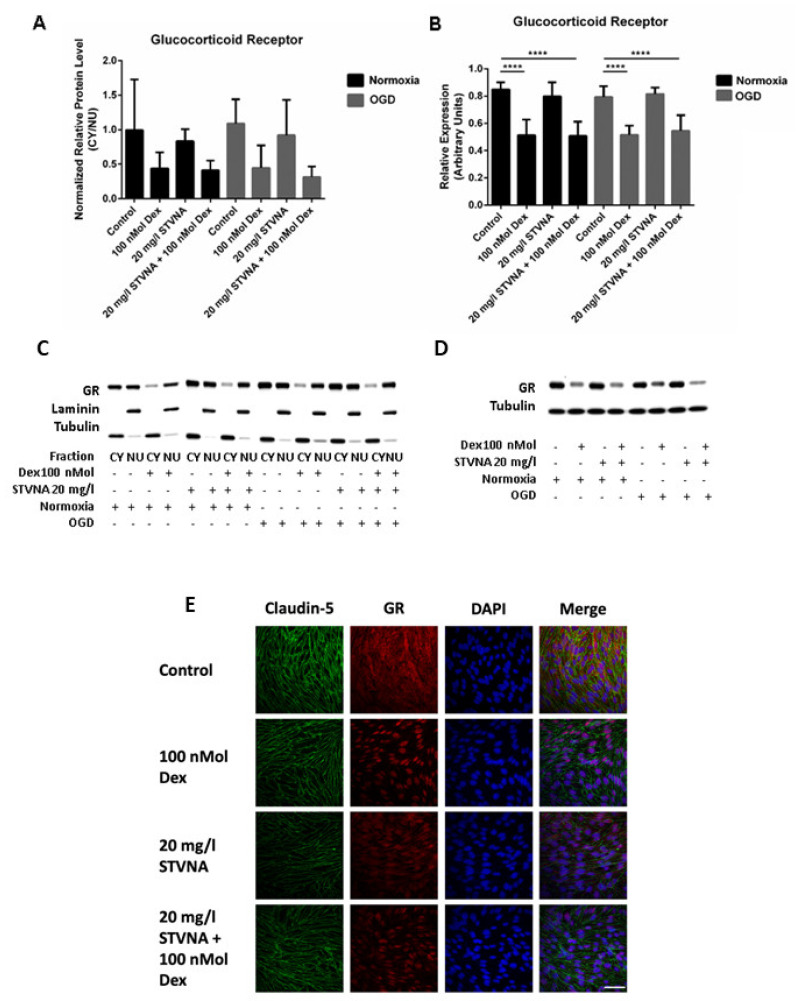
STVNA had no effects on transcriptional activation of GR. (**A**) Fractioned Western blot analysis of GR expression. Densitometric values were calculated relative to endogenous nuclear lamin A and cytoplasmic tubulin controls. CY = cytoplasmic fraction; NU = nuclear fraction. (**B**) Western blot of GR expression in whole cell lysates. **** *p* ≤ 0.0001. (**C**,**D**) Representatives blots of A and B. Blot shown is epresentative of *n* = 3 and *n* = 5 independent experiments for A and B, respectively. (**E**) Confocal immunofluorescence imaging of cells treated with STVNA post-OGD. GR= glucocorticoid receptor; nuclei stained with DAPI; staining of tight junctional protein claudin-5 served as a marker for cerebEND cells. Magnification = 60×, Scale bar = 50 µm.

**Figure 3 pharmaceutics-14-01753-f003:**
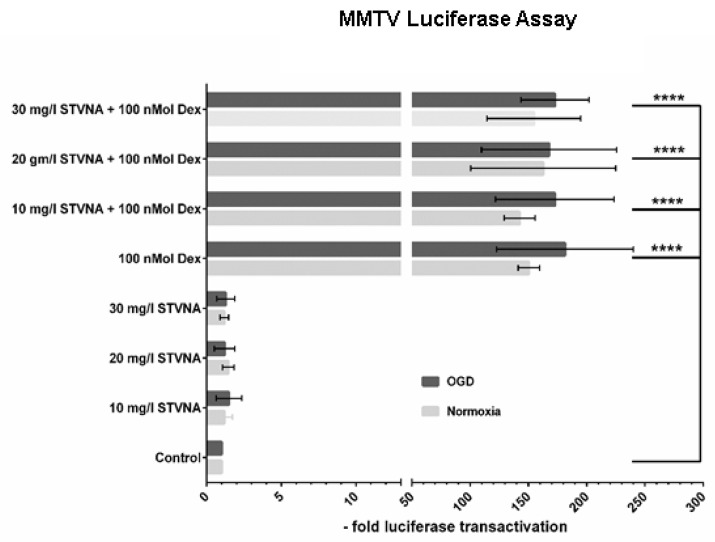
Glucocorticoid receptor (GR) activity remains the same after treatment with STVNA. Luciferase assay post-OGD, **** *p* ≤ 0.0001, *n* = 3.

**Figure 4 pharmaceutics-14-01753-f004:**
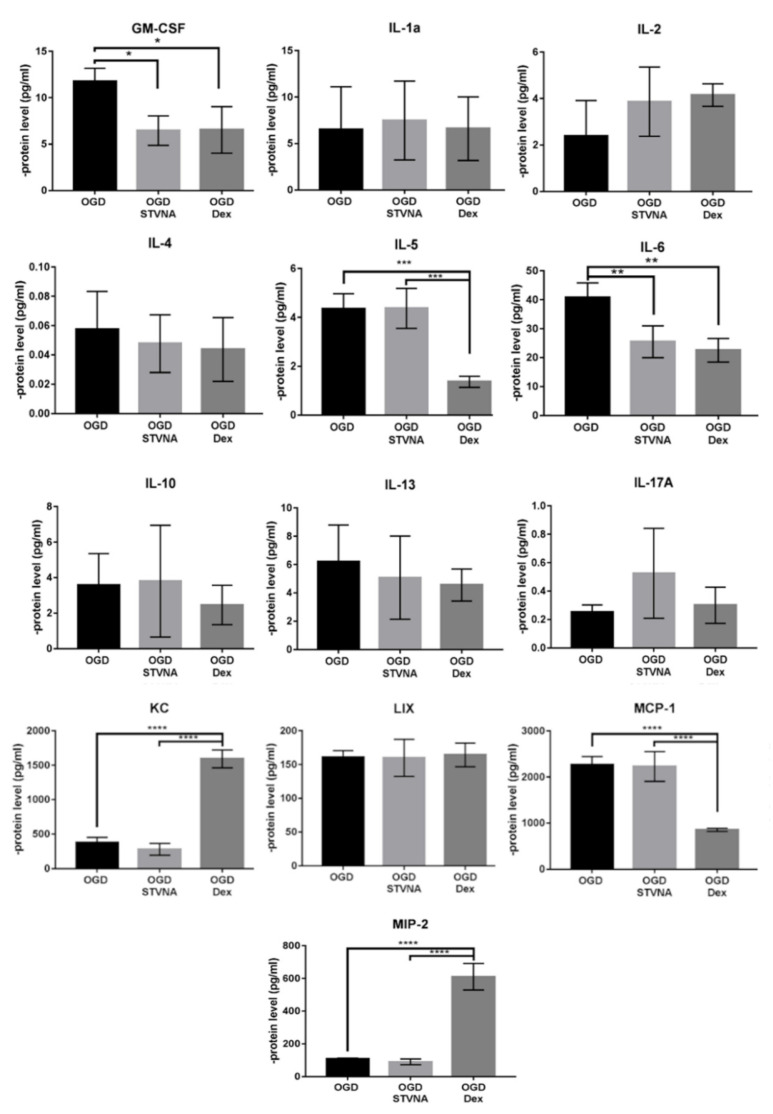
Multiplex assay of T-cell panel cytokines confirms reduction of OGD-induced IL-6 expression upon treatment with 20 mg/L STVNA, * *p* ≤ 0.05, ** *p* ≤ 0.01, *** *p* ≤ 0.001, **** *p* ≤ 0.0001, *n* = 4.

## Data Availability

The data presented in this study are available in the article text.

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
