# Peer review of "Isosteviol Sodium (STVNA) Reduces Pro-Inflammatory Cytokine IL-6 and GM-CSF in an In Vitro Murine Stroke Model of the Blood–Brain Barrier (BBB)"

_pharmaceutics, 2022, doi:10.3390/pharmaceutics14091753_

Round 1

Reviewer 1 Report

The article is well designed and written. The study seems interesting, it could be accepted due to  significance and further advantages compared to the existing literature about the same topic.

My suggestions:

1. The authors should describe the anti-inflammatory activity of STVNA assessed with other models (e.g. acetyl and butyl cholinesterase (AChE and BuChE) inhibitor activity.)on the basis of own research or available literature in this field.

2. The authors should describe blood brain barrier permeability by PAMPA BBB model on the basis of own research or available literature in this field.

Author Response

Dear Reviewer,

Thank you very much for your helpful feedback regarding our manuscript. Below, please find our point-by-point answers:

Comments and Suggestions for Authors

The article is well designed and written. The study seems interesting, it could be accepted due to  significance and further advantages compared to the existing literature about the same topic.

Thank you very much for your uplifting and motivating words.

My suggestions:

  1. The authors should describe the anti-inflammatory activity of STVNA assessed with other models (e.g. acetyl and butyl cholinesterase (AChE and BuChE) inhibitor activity.)on the basis of own research or available literature in this field. To our knowledge, there is no available data about the anti-inflammatory effects of STVNA regarding the AChE/BuChE-activity. Research by Zhang et al. (2017, doi: 10.1016/j.jstrokecerebrovasdis.2017.06.023; Sci Rep. 2019 Aug 21;9(1):12221. doi: 10.1038/s41598-019-48759-0.) indicated an inhibition via the NFkB signaling pathway. This reference is included in our introduction. Our own research also showed a minor downregulation of NFkB in some groups, but the data are not included as none of the effects were statistically significant.
  2. The authors should describe blood brain barrier permeability by PAMPA BBB model on the basis of own research or available literature in this field. Research about the BBB permeability has not been conducted by our group. Looking at the available literature we also could not find any data on this topic. Nonetheless, in our conclusions, we added as an outlook the necessity to investigate this.

Best regards,

Authors

Reviewer 2 Report

Some comments are listed here for the authors’ consideration to further improve the quality and overall impact of the manuscript.

Line 33 clarify what “GR” means.

Line 37 clarify what “GM-CSF” means.

Line 67-68 “…it was reported that dexamethasone, in combination with the proteasome inhibitor Bortezomib, reduced BBB permeability and brain edema.” Missing reference.

Line 72. Missing reference

At statistical analysis, authors should clarify what kind of ANOVA they used and the post hoc test.

Authors mention “Indeed, we could detect a trend towards reduced IL-6 expression in our murine microvascular cerebellar endothelial cells cerebEND in comparison to control upon administration of STVNA post-OGD.” But here, It is not clear, how the measures were done at cell cultures. Each condition (Normoxia pre, Normoxia post, ODG pre, ODG post) were doen at the same culture? or there was one per condition? And what ANOVA was used here? Figure 1.

At the same Figure 1, dexamethasone alone seems to decrease the IL-6 levels at Noxmoxia-post condition respect to control. What is this effect due to? Also, this effect is observed with the combination of STVNA and dexamethasone. Why is this effect observed here?

Also, dexamethasone alone seems to decrease the IL-6 levels at OGD-post condition, is this effect of same magnitude as observed with 20 mg/l STVNA?

Is the decrease of IL-6 levels at OGD-post condition with the combination of 20 mg/l STVNA and dexamethasone better than the administration of STVNA or dexamethasone alone?

At Figure 2A and 2B, the bars of the graphs do not differ in color, between both conditions Normoxia and OGD. Modify and better contrast colors.

For Figure 2B, mention at Figure captions, what statistical differences are indicated with * or **** and what experimental group correspond such difference.

Indicate the results of blots at Figure 2, as Figure 2C and 2D. And confocal immunofluorescence imaging of cells as Figure 2E.

The Figure 4, is small and the titles of axes are not clear. Modify the figure and letter size.

Also, for Figure 4, mention at Figure captions, what statistical differences are indicated with *, **, *** or **** and what experimental group correspond such difference.

Nothing is mentioned regarding the results of MIP-2, KC, MCP-1 OR IL-5, at results section.

At the conclusión, authors mention that “our present study suggests significant benefits of STVNA treatment (alone or in combination with dexamethasone) at the cellular level…”, but this statement is not true, because the effects observed in conbination with dexamethasone seems due to just dexamethasone effect.

Author Response

Dear Reviewer,

Thank you very much for your helpful and valuable feedback regarding our manuscript. Below, please find our point-by-point response to your comments and suggestions:

Comments and Suggestions for Authors

Some comments are listed here for the authors’ consideration to further improve the quality and overall impact of the manuscript.

Line 33 clarify what “GR” means.

The full terminology has been added.

Line 37 clarify what “GM-CSF” means.

The full terminology has been added.

Line 67-68 “…it was reported that dexamethasone, in combination with the proteasome inhibitor Bortezomib, reduced BBB permeability and brain edema.” Missing reference.

The reference has been added accordingly.

Line 72. Missing reference

The reference has been moved from the prior statements since they are the same sources.

At statistical analysis, authors should clarify what kind of ANOVA they used and the post hoc test.

We used one-way ANOVA with post-hoc Tukey´s multiple comparisons test. This has now been clarified in the text.

Authors mention “Indeed, we could detect a trend towards reduced IL-6 expression in our murine microvascular cerebellar endothelial cells cerebEND in comparison to control upon administration of STVNA post-OGD.” But here, It is not clear, how the measures were done at cell cultures. Each condition (Normoxia pre, Normoxia post, ODG pre, ODG post) were doen at the same culture? or there was one per condition? And what ANOVA was used here? Figure 1.

The same culture was divided into the four treatment groups for each replication. We clarified this now already in the text.

At the same Figure 1, dexamethasone alone seems to decrease the IL-6 levels at Noxmoxia-post condition respect to control. What is this effect due to? Also, this effect is observed with the combination of STVNA and dexamethasone. Why is this effect observed here?

Also, dexamethasone alone seems to decrease the IL-6 levels at OGD-post condition, is this effect of same magnitude as observed with 20 mg/l STVNA?

Is the decrease of IL-6 levels at OGD-post condition with the combination of 20 mg/l STVNA and dexamethasone better than the administration of STVNA or dexamethasone alone?

Dexamethasone is known to be an anti-inflammatory agent and decreases IL-6 levels. A possible explanation why the effect is mostly seen in the post groups, is the time-dependence of dexamethasone effects. This theory is supported by reference 45 (Sun et al., 2018. Brain Res 1701:237-245).

At Figure 2A and 2B, the bars of the graphs do not differ in color, between both conditions Normoxia and OGD. Modify and better contrast colors.

Figure 2 has now been altered, made larger, with normoxia group in black column bars and OGD group depicted in gray color bars.

For Figure 2B, mention at Figure captions, what statistical differences are indicated with * or **** and what experimental group correspond such difference.

**** P ≤ 0.0001. This has been clarified in the figure legend.

Indicate the results of blots at Figure 2, as Figure 2C and 2D. And confocal immunofluorescence imaging of cells as Figure 2E.

This has been corrected as advised.

The Figure 4, is small and the titles of axes are not clear. Modify the figure and letter size.

The figure has been modified.

Also, for Figure 4, mention at Figure captions, what statistical differences are indicated with *, **, *** or **** and what experimental group correspond such difference.

**** P ≤ 0.0001. This has been clarified in the figure legend.

Nothing is mentioned regarding the results of MIP-2, KC, MCP-1 OR IL-5, at results section.

The results for the aforementioned are now included in the text of the results section.

At the conclusión, authors mention that “our present study suggests significant benefits of STVNA treatment (alone or in combination with dexamethasone) at the cellular level…”, but this statement is not true, because the effects observed in conbination with dexamethasone seems due to just dexamethasone effect.

We modified our statement for more clarity by exchanging a generalized one into a more specific conclusion.

Kindest regards,

Authors

Reviewer 3 Report

Isosteviol Sodium (STVNA) reduces pro-inflammatory cytokine IL-6 and GM-CSF in an in vitro 1 murine stroke model of the blood brain barrier (BBB) by Moritz Reschke et al.

Abstract

The authors tested whether STVNA can activate glucocorticoid receptor (GR) transcriptional activity in brain microvascular endothelial cells. STVNA did not activate GR as showed by luciferase assay. But STVNA inhibited IL-6 and GM-CSF secretion after OGD conditions modeling in vitro stroke.

The manuscript is easy to read, although it is confusing at times. The presented data are also incomplete and confusing at times. More data is needed to evaluate the effects of STVNA on the in vitro BBB model. After major correction this manuscript may be suitable for a rapid communication.

Comments and suggestions

Abstract

Line 33: please write full name for GR at first mention

Please include in the abstract OGD, it is not mentioned anywhere. IL-6 and GM-CSF increase occurs in OGD conditions. 

Methods

Line 112: "Pre-treatment groups were treated with 20mg/l STVNA...." Several concentrations are shown on Figure 1.

Results

Line 190: "STVNA reduces OGD-induced IL-6 expression"

Figure 1: This statement has to be supported by data. Please mark the statistical significance between the groups on the graph (Figure 1).

Figure 2A: There is no STVNA treatment marked under the blots in Normoxia but appears in the graph above. Please correct it. The western blots shown for cytoplasmic/nuclear fraction do not have all the groups to create Fig 2A!

Looking at the western blots, it appears that the cytoplasmic GR decreases but the nuclear levels do not change much.

Please align GR, Laminin and Tubulin labeling with their western blots.

Line 223: "..confocal immunofluorescence microscopy revealed that no definite nuclear translocation of GR..." On the presented image (Figure 2C) some nuclear translocation of GR is visible similar to Dex treatment. Please mention this also.

Line 227: Figure 2 legend: "STVNA had no effects on transcriptional activation of GR." The confocal images do not support this statement. Please comment on this in the Discussion.

Line 233: "Staining of tight junctional protein claudin-5 served as a marker for cerebEND cells."

Line 253: "Decreased IL-6 expression after co-administration of dexamethasone and STVNA treatment" based on the data, STVNA was able to decrease IL-6 on its own. Dex also decreased it alone. There is no co-treatment with STVNA+Dex presented on the graphs!

Line 276: "Its capability to produce the same effects in cerebral endothelial cells..."

Line 310: "which results in..."

Line 329: "By and large, STVNA could be an interesting treatment option due to the effects it is able to generate independent of GR."  Based on Figure 2C, GR may partially translocate to the nucleus, even though no change in GR activity is detected with the luciferase assay. Please comment briefly on this issue.

Line 333: "Still, our present study suggests significant benefits of STVNA treatment (alone or in combination with dexamethasone)...."  Based on the presented data, STVNA does not augment the dexamethasone effect (no additive or synergistic effects are observed).

Line 275: "The ability of STVNA to reduce IL-6 levels was already displayed in neuroblastoma cells (Zhong et al., 2018). Its capability to produce the same effects in cerebellar endothelial cells indicates the feasibility of it reducing the negative effects associated with BBB disruption during brain ischemia-reperfusion injury."

Because IL-6 decrease by STVNA was reported in another CNS cell type, the authors should additionally focus on how STVNA is affecting the BBB characteristics of their CerebEND cells (TJ protein expression, localization, perhaps permeability changes, if this cell line is suitable for permeability studies). The authors showed nice claudin-5 staining in their cells but they considered claudin-5 as an internal control and did not show changes of this tight junction protein. Based on the images from Figure 2C, it seems that STVNA is decreasing claudin-5 immunoreactivity. Some higher magnification images would help to see the changes better.

Special Issue: Women in Pharmaceutics

I do not agree with the mission statement of this Special issue.

Women scientists nowadays can achieve as much as their male counterparts. There are countless opportunities and examples of this.  My early mentor was a very successful female scientist and our department chair is also an excellent woman scientist. I encounter many women in leading positions daily. Based on my personal experience, no women is discriminated against in scientific forums. These forums are open and welcoming for everyone.

Scientific publications should not consider the scientist's gender. They should solely focus on the scientific merit of the work. We have to respect women and trust them enough that they are able to achieve excellence in science. Every woman should advance their career based on her knowledge, performance and character, not based on her gender. Thank you for considering these important aspects.

Author Response

Dear Reviewer,

We are grateful for your valuabla and helpful comments and suggestions. Below, please find our point-by-point answers to those comments.

Sincerely yours,

Authors

Comments and Suggestions for Authors

Isosteviol Sodium (STVNA) reduces pro-inflammatory cytokine IL-6 and GM-CSF in an in vitro 1 murine stroke model of the blood brain barrier (BBB) by Moritz Reschke et al.

Abstract

The authors tested whether STVNA can activate glucocorticoid receptor (GR) transcriptional activity in brain microvascular endothelial cells. STVNA did not activate GR as showed by luciferase assay. But STVNA inhibited IL-6 and GM-CSF secretion after OGD conditions modeling in vitro stroke.

The manuscript is easy to read, although it is confusing at times. The presented data are also incomplete and confusing at times. More data is needed to evaluate the effects of STVNA on the in vitro BBB model. After major correction this manuscript may be suitable for a rapid communication.

Thank you very much for your motivating statement. We also are aware that our data need a follow-up investigation and we are herein only reporting preliminary albeit what we deem to be significant results.

Comments and suggestions

Abstract

Line 33: please write full name for GR at first mention

This has now been done.

Please include in the abstract OGD, it is not mentioned anywhere. IL-6 and GM-CSF increase occurs in OGD conditions.

We have included as advised.

Methods

Line 112: "Pre-treatment groups were treated with 20mg/l STVNA...." Several concentrations are shown on Figure 1.

Thank you very much for pointing out our oversight. This has now been amended in the text.

Results

Line 190: "STVNA reduces OGD-induced IL-6 expression"

We have modified this statement to match the data more appropriately.

Figure 1: This statement has to be supported by data. Please mark the statistical significance between the groups on the graph (Figure 1).

The statement in the figure legend has been modified to more appropriately match the data shown. No statistical

Figure 2A: There is no STVNA treatment marked under the blots in Normoxia but appears in the graph above. Please correct it. The western blots shown for cytoplasmic/nuclear fraction do not have all the groups to create Fig 2A!

The blot shown initially was indeed incomplete. We have now rectified this; as we have had an oversight when generating the figure.

Looking at the western blots, it appears that the cytoplasmic GR decreases but the nuclear levels do not change much.

Indeed, the nuclear levels do not change much, as we have reported. This tendency shows that GR activity, which could mostly be seen in the nuclear fraction, is unaltered.

Please align GR, Laminin and Tubulin labeling with their western blots.

This has been modified as advised.

Line 223: "..confocal immunofluorescence microscopy revealed that no definite nuclear translocation of GR..." On the presented image (Figure 2C) some nuclear translocation of GR is visible similar to Dex treatment. Please mention this also.

This suggestion has been taken and the statement has been added.

Line 227: Figure 2 legend: "STVNA had no effects on transcriptional activation of GR." The confocal images do not support this statement. Please comment on this in the Discussion.

We have added the following statement in the discussion: “Even though we detected a slight nuclear translocation of the GR after treatment with 20 mg/l STVNA in the immunofluorescence images, the activation of the GR was ruled out by the negative results in the luciferase assay.”

Line 233: "Staining of tight junctional protein claudin-5 served as a marker for cerebEND cells."

This has been modified.

Line 253: "Decreased IL-6 expression after co-administration of dexamethasone and STVNA treatment" based on the data, STVNA was able to decrease IL-6 on its own. Dex also decreased it alone. There is no co-treatment with STVNA+Dex presented on the graphs!

The heading has been modified to match the data more appropriately.

Line 276: "Its capability to produce the same effects in cerebral endothelial cells..."

We used cerebellar endothelial cells, not cerebral.

Line 310: "which results in..."

This was altered.

Line 329: "By and large, STVNA could be an interesting treatment option due to the effects it is able to generate independent of GR."  Based on Figure 2C, GR may partially translocate to the nucleus, even though no change in GR activity is detected with the luciferase assay. Please comment briefly on this issue.

We have added a statement in the discussion (Please see lines 309-311).

Line 333: "Still, our present study suggests significant benefits of STVNA treatment (alone or in combination with dexamethasone)...."  Based on the presented data, STVNA does not augment the dexamethasone effect (no additive or synergistic effects are observed).

This statement has been modified.

Line 275: "The ability of STVNA to reduce IL-6 levels was already displayed in neuroblastoma cells (Zhong et al., 2018). Its capability to produce the same effects in cerebellar endothelial cells indicates the feasibility of it reducing the negative effects associated with BBB disruption during brain ischemia-reperfusion injury."

We emphasised on the barrier properties of the STVNA effect in a previous publication from our work group (Rosing et al., 2020). In this manuscript, the emphasis was supposed to be on inflammation and GR.

Because IL-6 decrease by STVNA was reported in another CNS cell type, the authors should additionally focus on how STVNA is affecting the BBB characteristics of their CerebEND cells (TJ protein expression, localization, perhaps permeability changes, if this cell line is suitable for permeability studies). The authors showed nice claudin-5 staining in their cells but they considered claudin-5 as an internal control and did not show changes of this tight junction protein. Based on the images from Figure 2C, it seems that STVNA is decreasing claudin-5 immunoreactivity. Some higher magnification images would help to see the changes better.

We have modified this figure.

 Special Issue: Women in Pharmaceutics

I do not agree with the mission statement of this Special issue.

Women scientists nowadays can achieve as much as their male counterparts. There are countless opportunities and examples of this.  My early mentor was a very successful female scientist and our department chair is also an excellent woman scientist. I encounter many women in leading positions daily. Based on my personal experience, no women is discriminated against in scientific forums. These forums are open and welcoming for everyone.

Scientific publications should not consider the scientist's gender. They should solely focus on the scientific merit of the work. We have to respect women and trust them enough that they are able to achieve excellence in science. Every woman should advance their career based on her knowledge, performance and character, not based on her gender. Thank you for considering these important aspects.

Round 2

Reviewer 3 Report

The authors addressed all questions and concerns properly except for this minor issue:

Line 200: "The administration of 20 mg/l STVNA decreased IL-6 secretion, both alone and in combination with dexamethasone." Please mark statistical significance on the graph or, if not statistically significant, change the text from "decreased" to "decreasing trend".

This manuscript is a resubmission of an earlier submission. The following is a list of the peer review reports and author responses from that submission.

Round 1

Reviewer 1 Report

The authors taken into account almost all the reviewers suggestions. It remains sentences like:

"This nuclear translocation was enhanced after co-treatment with dexamethasone, although results were not significant (Figure 2A)"

that should be removed from the final text !

A result is significant or not !

Author Response

The authors have taken the reviewer´s comment into consideration and changed the text accordingly.

Reviewer 2 Report

Quality of the manuscript has been improved by adding regulation data of GM-CSF and other cytokines and chemokines. My concern about the relevance of the cellular model for blood brain barrier is no longer relevant due to the change of the focus of the manuscript. Though STVNA could be an interesting treatment option due to the independence on GR, the authors should discuss also the limitation of this potential treatment due to lack of efficacy on most of the chemokines and cytokines tested, whereas  the effects of dexamethasone are broader.

Author Response

It is not our aim to show that STVNA could be a replacement for dexamethasone. Rather, we wanted to show a comparable effect of STVNA in this particular case. Nonetheless, the comment and suggestion of the reviewer has been taken into account and we have included the limitation of STVNA as potential treatment option in the discussion.